# Factors contributing to healthcare professional burnout during the COVID-19 pandemic: A rapid turnaround global survey

**Luca A. Morgantini**[1]*, **Ushasi Naha**[1], **Heng Wang**[2], **Simone Francavilla**[1], **Ömer Acar**[1], **Jose M. Flores**[1], **Simone Crivellaro**[1], **Daniel Moreira**[1], **Michael Abern**[1], **Martin Eklund**[3], **Hari T. Vigneswaran**[1,3], **Stevan M. Weine**[4,5]

1 Department of Urology, College of Medicine, University of Illinois at Chicago, Chicago, Illinois, United States of America, 2 Department of Epidemiology and Biostatistics, School of Public Health, University of Illinois at Chicago, Chicago, Illinois, United States of America, 3 Department of Medical Epidemiology and Biostatistics, Karolinska Institute, Stockholm, Sweden, 4 Department of Psychiatry, College of Medicine, University of Illinois at Chicago, Chicago, Illinois, United States of America, 5 Center for Global Health, University of Illinois at Chicago, Chicago, Illinois, United States of America

* lmorga5@uic.edu

**Data Availability Statement:** All relevant data are within the manuscript and its Supporting Information files.

## Abstract

### Background

Healthcare professionals (HCPs) on the front lines against COVID-19 may face increased workload and stress. Understanding HCPs' risk for burnout is critical to supporting HCPs and maintaining the quality of healthcare during the pandemic.

### Methods

To assess exposure, perceptions, workload, and possible burnout of HCPs during the COVID-19 pandemic we conducted a cross-sectional survey. The main outcomes and measures were HCPs' self-assessment of burnout, indicated by a single item measure of emotional exhaustion, and other experiences and attitudes associated with working during the COVID-19 pandemic.

### Findings

A total of 2,707 HCPs from 60 countries participated in this study. Fifty-one percent of HCPs reported burnout. Burnout was associated with work impacting household activities (RR = 1·57, 95% CI = 1·39–1·78, *P*<0·001), feeling pushed beyond training (RR = 1·32, 95% CI = 1·20–1·47, *P*<0·001), exposure to COVID-19 patients (RR = 1·18, 95% CI = 1·05–1·32, *P* = 0·005), and making life prioritizing decisions (RR = 1·16, 95% CI = 1·02–1·31, *P* = 0·03). Adequate personal protective equipment (PPE) was protective against burnout (RR = 0·88, 95% CI = 0·79–0·97, *P* = 0·01). Burnout was higher in high-income countries (HICs) compared to low- and middle-income countries (LMICs) (RR = 1·18; 95% CI = 1·02–1·36, *P* = 0·018).

**Funding:** The authors received no specific funding for this work.

**Competing interests:** The authors have declared that no competing interests exist.

## Interpretation

Burnout is present at higher than previously reported rates among HCPs working during the COVID-19 pandemic and is related to high workload, job stress, and time pressure, and limited organizational support. Current and future burnout among HCPs could be mitigated by actions from healthcare institutions and other governmental and non-governmental stakeholders aimed at potentially modifiable factors, including providing additional training, organizational support, and support for family, PPE, and mental health resources.

## Introduction

More than 200 countries worldwide are impacted by the spread of the novel coronavirus (SARS-Cov-2), the pathogen responsible for the coronavirus disease 2019 (COVID-19). Their healthcare systems are frantically maximizing efforts to deploy resources in order to mitigate spread and reduce morbidity and mortality from COVID-19.

Large numbers of healthcare professionals (HCPs) on the frontlines face high adversity, workloads, and stress, making them vulnerable to burnout [1, 2]. Burnout, defined by emotional exhaustion, depersonalization, and personal accomplishment, is known to detract from optimal working capacities, and has been previously shown to be similarly prevalent among HCPs in HICs (High-Income Countries) and LMICs (Low-to-Middle-Income Countries) [3–6]. Burnout has been found to be driven by high job stress, high time pressure and workload, and poor organizational support. These factors are common between HICs and LMICs despite their differences in healthcare and socioeconomic structures [3].

Researchers have begun exploring the impact of the COVID-19 pandemic on HCPs' mental health. Barello et al. assessed 376 Italian HCPs who interacted with COVID-19 infected patients for their reported burnout, psychosomatic symptoms and self-perceived general health, finding in their study population high emotional burnout, physical symptoms, and work-related pressure [7]. The Society of Critical Care Medicine surveyed 9492 intensive care unit clinicians in the U.S. and found that median self-reported stress, measured on a scale from 0 to 10, increased from 3 to 8 during the pandemic [8]. The principal stressors included concern for lack of personal protective equipment (PPE), and work impacting household activities and interactions [8]. Shanafelt et al. identified the necessity for HCPs to care for patients that required clinical skills beyond their training as an additional stressor, among others [9]. The pandemic has not affected all HCPs in the same manner, as there have been demonstrated differences based on occupation and patient population. Lai et al. demonstrated how HCPs in Wuhan, especially nurses and frontline workers, were experiencing the highest psychological burden in late January 2020 [1]. Zerbini et al. identified how German nurses working in COVID-19 wards reported worse burnout scores compared to their colleagues in regular wards, while physicians reported similar scores independently from their COVID-19 workload [10]. In contrast, Wu et al. reported in their study of 190 HCPs in Wuhan how individuals working in their usual ward reported a higher frequency of burnout and fear of being infected, when compared to their colleagues working with COVID-19 patients [11]. Differences in the perception of the pandemic, the local spread of the pandemic at the time of study, support structures, or definition of burnout that may explain these diametrically opposed results [12]. Because each study focused on HCPs working in a particular region or country, it is impossible to draw conclusions about the impact on HCPs globally.

The objective of this study was to understand the impact of COVID-19 on HCPs around the world working during the pandemic. This was the first intercontinental survey examining the perceptions of HCPs during the COVID-19 pandemic without restriction on geographic location or COVID-19 exposure. Given that the pandemic has not affected all nations in the same time frame, gathering opinions from HCPs worldwide within a single time range offers a unique snapshot of how the pandemic affects HCPs at that moment. Our aim was to describe current contributing factors associated with HCPs burnout during the pandemic and to provide data that will drive future research on mitigating burnout.

## Methods

### Human subjects research

The University of Illinois at Chicago (UIC) Institutional Review Board (IRB) determined on April 1st, 2020 that this study, with assigned protocol number 2020–0388, met the criteria for exemption as defined in the U.S. department of Health and Human Services Regulations for the Protection of Human Subjects [45 CFR 46. 104(d)]. Before initiating the survey, respondents were informed that their responses would be shared with the scientific community. Survey responses were recorded and stored without participant identifiers using the REDCap electronic data capture software hosted by UIC servers [13, 14]. REDCap (Research Electronic Data Capture) is a secure, web-based software platform designed to support data capture for research studies, providing 1) an intuitive interface for validated data capture; 2) audit trails for tracking data manipulation and export procedures; 3) automated export procedures for seamless data downloads to common statistical packages; and 4) procedures for data integration and interoperability with external sources.

### Sample population and recruitment strategy

Inclusion criteria was restricted to HCPs. Platforms including Facebook, WhatsApp, and Twitter, as well as e-mail, were used for global recruitment and dissemination from April 6 to April 16, 2020. Potential study participants were approached via IRB-approved messages containing a link to the survey shared on the aforementioned social media. Study participants were also asked to share the link with their colleagues via personal networks.

### Outcomes and measures

Demographic data collected from the survey participants was limited to the country of provenience and occupation. The survey contained 40 questions covering three major domains of HCPs experience (exposure, perception, and workload) that were validated by experts in infectious diseases, public health, occupational medicine, psychology, and clinical psychiatry. Elements of these domains were previously proposed as contributing toward HCP anxiety during the COVID-19 pandemic [15]. The main outcome, HCPs-perceived burnout in its core domain of emotional exhaustion, was assessed by a single item on a 7-point Likert scale (1: strongly disagree to 7: strongly agree) using the statement, "I am burned out from my work [16]. Only the core domain of emotional exhaustion was assessed as previous research has demonstrated that the depersonalization and personal accomplishment domains represented a Western concept not generalizable across different cultures [12].

The questionnaire was developed with a pilot group of 10 HCPs and 40 questions were included based on expert opinion (S1 Questionnaire) that were then translated into 18 languages by professional translators. The country of the respondents was categorized as high-income or low- and middle-income as defined by the World Bank classification system in

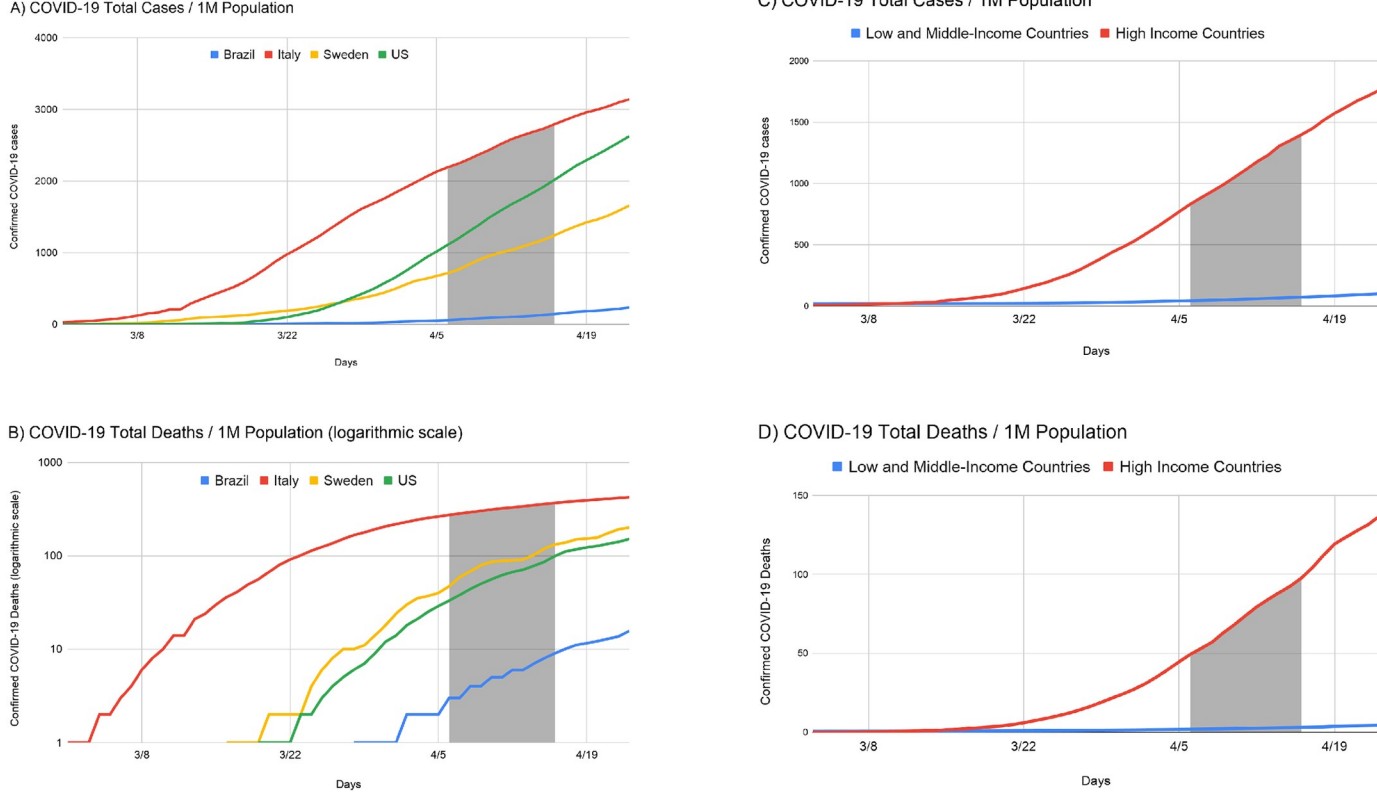

**Fig 1. Total confirmed COVID-19 cases (A) and total confirmed COVID-19 deaths (B) per 1 million (M) population for the 4 countries with the highest response rates and for HICs (C) and LMICs (D).**

order to reduce confounding factors such as differences in the number of COVID-19 cases (Fig 1), healthcare system, and socioeconomic structure [3, 17]. COVID-19 deaths and cases per 1 million population were obtained from a widely used web-based dashboard [18].

## Statistical analyses

A descriptive assessment was performed for each variable surveyed for all data, country by country, and according to the income level (high vs. low-middle). Covariates collected as ordinal variables were transformed into binary (S1 Table). For burnout, scores $\geq 5$ were considered burned out [16]. Quasi-Poisson regression analysis was performed using the binary burnout outcome to compare factors associated with low and average burnout against high emotional exhaustion burnout [19]. Relative risk (RR) was reported with nominal 95% confidence intervals and 2-sided P values. Only the participants who responded completely to the variables of interest were included in regression analyses.

## Results

A total of 2,707 responses were received from HCPs in 60 countries. Fig 1 demonstrates the study period in context of the COVID-19 pandemic (S2 Table) [17, 18].

Table 1 summarizes participant characteristics and responses (additional responses in S3 Table). Half (51·4%) of the respondents from 33 countries reported emotional exhaustion

**Table 1. Healthcare professionals' responses about perceptions, exposure, and workload during the COVID-19 pandemic.**

| Country | |
|---|---|
| Brazil | 186 (6·9%) |
| Italy | 598 (22·1%) |
| USA | 833 (30·8%) |
| Sweden | 149 (5·5%) |
| Other | 941 (34·8%) |
| **Country level of income** | |
| Low-to-Middle-Income Countries (LMIC) | 314 (19%) |
| High-Income Countries (HIC) | 1334 (81%) |
| **Occupation category** | |
| Physician (Residents, Fellows) | 719 (26·6%) |
| Nurse (NP, PA, CRNA) | 855 (31·6%) |
| Other | 1133 (41·9%) |
| **Exposed to a patient with COVID-19** | |
| No | 644 (33·9%) |
| Yes | 1255 (66·1%) |
| **Symptoms suggestive of COVID-19** | |
| No | 1526 (80·2%) |
| Yes | 377 (19·8%) |
| **Tested for COVID-19** | |
| No | 1630 (85·7%) |
| Yes | 271 (14·3%) |
| **Positive test for COVID-19** | |
| No | 221 (83·1%) |
| Yes | 45 (16·9%) |
| **Current perception of COVID-19** | |
| Benign disease | 16 (0·9%) |
| Mild disease | 50 (2·9%) |
| Moderate disease | 534 (30·8%) |
| Severe disease | 1134 (65·4%) |
| **Adequate PPE was provided** | |
| No | 778 (45·2%) |
| Yes | 945 (54·8%) |
| **Was mental health support available** | |
| No | 902 (52·2%) |
| Yes | 825 (47·8%) |
| **Received COVID-19-specific training** | |
| No | 921 (53·1%) |
| Yes | 815 (46·9%) |
| **Made life prioritizing decision** | |
| No | 1470 (85·6%) |
| Yes | 248 (14·4%) |
| **Felt pushed beyond training** | |
| No | 1174 (68·1%) |
| Yes | 550 (31·9%) |
| **Work impacting household activities because of COVID-19** | |
| No | 500 (30·5%) |

*(Continued)*

**Table 1.** (Continued)

| Country | |
|---|---|
| **Yes** | 1139 (69·5%) |
| **Work impacting QoL because of COVID-19** | |
| **No** | 538 (32·8%) |
| **Yes** | 1100 (67·2%) |
| **I am burned out from my work (Likert 1–7)** | |
| **Strongly disagree** | 146 (8·9%) |
| **Disagree** | 255 (15·6%) |
| **Somewhat disagree** | 114 (7·0%) |
| **Neither agree nor disagree** | 281 (17·2%) |
| **Somewhat agree** | 406 (24·8%) |
| **Agree** | 249 (15·2%) |
| **Strongly agree** | 187 (11·4%) |
| **I am burned out from my work (Binary)** | |
| **No** | 796 (48·6%) |
| **Yes** | 842 (51·4%) |

(PPE) Personal protective equipment; (QoL) Quality of life; (NP) Nurse practitioner; (PA) Physician assistant; (CRNA) Certified registered nurse anesthetist.

burnout related to their work during the COVID-19 pandemic. The U.S. had the highest reported burnout among all countries at a rate of 62·8%.

Across all countries (Fig 2), in the multivariable regression analysis, reported burnout was associated with work impacting household activities (RR = 1·57, 95% CI = 1·39–1·78, $P<0·001$), feeling pushed beyond training (RR = 1·32, 95% CI = 1·20–1·47, $P<0·001$), exposure to COVID-19 patients (RR = 1·18, 95% CI = 1·05–1·32, $P = 0·005$), and making life prioritizing decisions due to supply shortages (RR = 1·16, 95% CI = 1·02–1·31, $P = 0·03$). Adequate PPE was protective against reported burnout (RR = 0·88, 95% CI = 0·79–0·97, $P = 0·01$). The answers of the individuals that were not included in the regression analyses due to missing data did not significantly differ from those who did completely respond.

Country-level analysis revealed lower reported burnout in Italy (RR = 0·72, 95% CI = 0·61–0·84, $P<0·001$) and Sweden (RR = 0·43, 95% CI = 0·30–0·59, $P<0·001$) compared to the U.S.

Predictors of burnout differed between LMICs and HICs (S1 Fig). Among the 314 respondents from LMICs, reported burnout was associated with work impacting household activities (RR = 2·31, 95% CI = 1·61–3·43, $P<0·001$) and adequate PPE (RR = 0·68, 95% CI = 0·52–0·90, $P = 0·007$). In the 1334 respondents from HICs, reported burnout was associated with feeling pushed beyond training (RR = 1·41, 95% CI = 1·06–1·88, $P = 0·02$), difficulty obtaining COVID-19 testing (RR = 1·43, 95% CI = 1·04–1·94, $P = 0·03$), work impacting quality of life (RR = 1·67, 95% CI = 1·12–2·59, $P = 0·02$), work impacting household activities (RR = 1·75, 95% CI = 1·16–2·75, $P = 0·01$), and mental health support (RR = 0·72, 95% CI = 0·54–0·96, $P = 0·03$).

## Discussion

Among respondents, half of HCPs from 33 countries reported burnout. Previously reported rates of HCP burnout have ranged from 43% to 48% [3]. Burnout for HCPs working during the COVID-19 pandemic was associated with factors that typically increase the likelihood of HCP burnout [1, 3]. These included feeling pushed beyond training (high workload), making

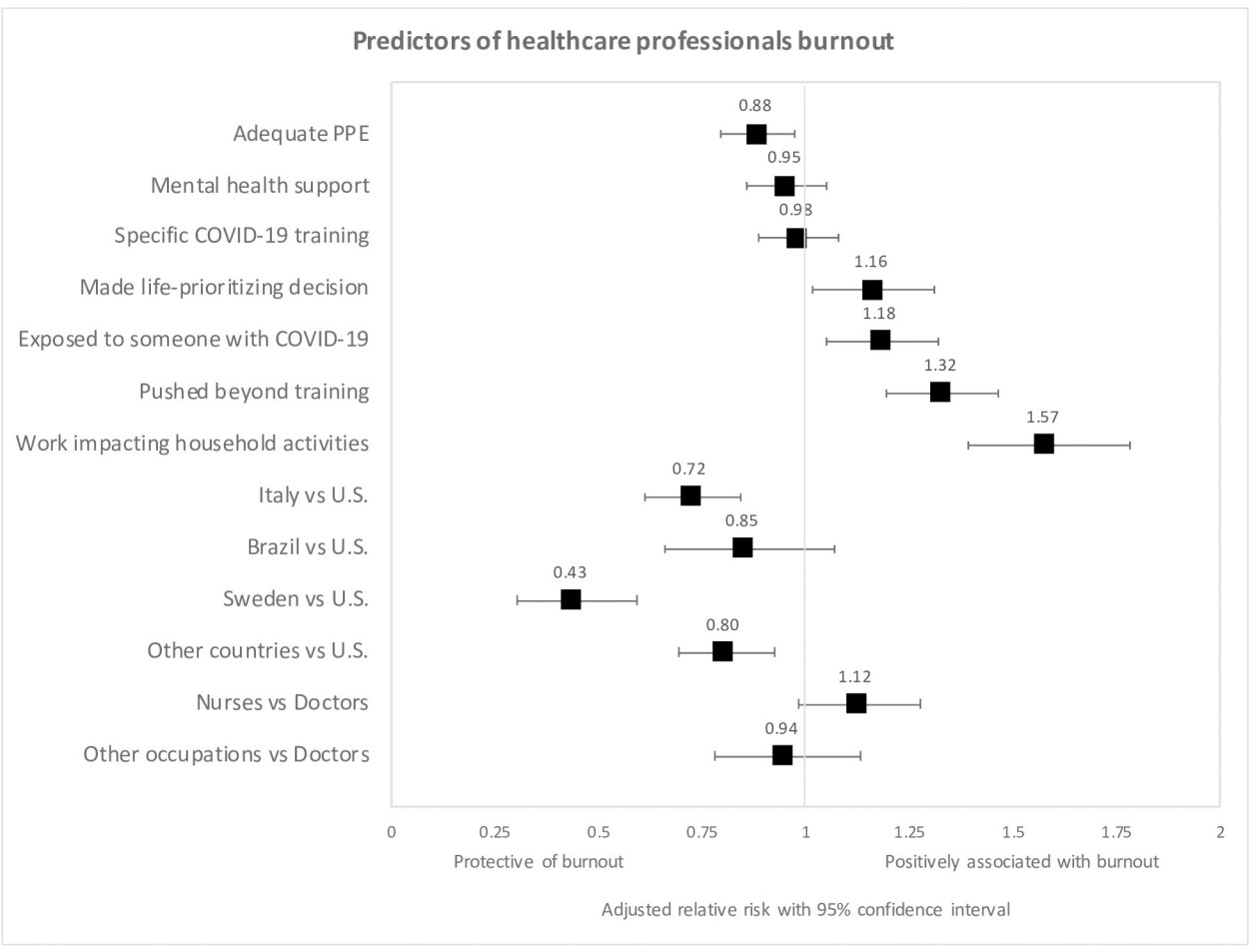

**Fig 2. Forest plots show adjusted Relative Risk (RR) for the multivariable regression analysis of burnout.** (PPE) Personal protective equipment; (ICU) Intensive care unit; (ER) Emergency room; (ID) Infectious diseases.

life-or-death prioritizing decisions (high job stress), work impacting the ability to perform household activities (high time pressure), and lack of adequate PPE (limited organizational support).

Burnout among HCPs could be reduced by actions from healthcare institutions and other governmental and non-governmental stakeholders aimed at potentially modifiable factors. These could include providing additional training and mental health resources, strengthening organizational support for HCPs' physical and emotional needs, supporting family-related issues (e.g. helping with childcare, transportation, temporary housing, wages), and acquiring PPE. A systematic review showed that both individual- and organizational-level strategies are effective in meaningfully reducing burnout. Some of the most commonly utilized methods focused on mindfulness, stress management and small group discussion [20]. Future studies should examine if and how the implementation of such strategies can reduce burnout among HCPs during the pandemic.

Recent studies regarding HCPs' mental health in response to COVID-19 from China, as well as prior studies of other pandemics, have demonstrated that HCPs may experience depression, anxiety, and posttraumatic stress disorder. Shanafelt *et al.* highlighted common sources of anxiety from listening sessions with HCPs that align with our findings, such as

access to adequate PPE, unknowingly bringing the infection home, and lack of access to up-to-date information and communication [9]. Some HCPs who worked extensively during the SARS pandemic in Beijing later demonstrated posttraumatic stress symptoms (PTSS), and many HCPs in the areas hardest-hit by COVID-19 in China have already started exhibiting similar complaints [21, 22]. To prevent adverse psychological outcomes, mental health support for HCPs is critical [2, 23]. Key interventions include access to psychosocial support including web-based resources, emotional support hotlines, psychological first aid, and self-care strategies.

Burnout can impact not only mental health but also can correlate with physical ailments. A systematic review found that burnout was a predictor for conditions including musculoskeletal pain, prolonged fatigue, headaches, gastrointestinal and respiratory issues [24]. Some factors included in our survey, such as increased workload hours, inadequate PPE or lack of updated guidelines, contributed to higher rates of infection among HCPs at the beginning of the outbreak in late January [25].

Burnout was higher in those countries where the COVID-19 pandemic was surging at the time of data collection (e.g. the U.S.) compared with those where it was declining (e.g. Italy) or had not reached the peak (e.g. Turkey). The lower reported burnout among HCPs in LMICs may reflect resilience due to more experience working in conditions with high adversity and limited availability of supplies [26]. Additionally, the greater reported burnout by HCPs in HICs could be attributed to their greater COVID-19 burden. Addressing burnout in all countries is crucial, but our findings indicate that different strategies should be tailored to the phase of pandemic and the sociocultural and healthcare organizational contexts.

## Limitations

Despite this study's major strengths, including the breadth of responses from across the globe, there are multiple limitations including a non-validated questionnaire, not providing the definition of burnout to participants before the initiation of study, a single item indicator for burnout, minimal demographic data collection, and sampling method using social media. By utilizing recruitment and dissemination strategies dependent on social media, there is a potential selection bias resulting in overrepresentation of HCPs more active on social media forums. The lack of extensive demographic collection, designed to increase participation, limits the ability to assess the representativeness of the study sample.

Future studies should consider expanding beyond the single item to explain the complexity of burnout in HCPs as this study only represents an indication of reported emotional exhaustion in the study participants [16]. Causality between the COVID-19 pandemic and HCPs' reported burnout cannot be determined due to the nature of this observational study, and additional studies are needed to ascertain causality.

Furthermore, drawing comparisons between countries is limited by the differences in cultures, languages, and healthcare systems. The definition and perception of burnout varies across countries, although the domain of emotional exhaustion, as investigated in this study, remains consistent and validated across all translated languages [12].

## Conclusions

While HCPs wage a war against COVID-19, institutions must support these individuals as they face enormous stress that can negatively impact their emotional and physical well-being. Our study is the first worldwide survey of HCPs during the COVID-19 pandemic and demonstrates the presence of reported burnout among respondents at a rate higher than previously reported. Reported burnout was significantly associated with, among others, limited access to

PPE as well as making life-or-death decisions due to medical supply shortages. Furthermore, reported burnout was associated with different factors in HICs and LMICs. Current and future burnout among HCPs could be mitigated by actions from healthcare institutions and other governmental and non-governmental stakeholders aimed at potentially modifiable factors, including providing additional training, organizational support, support for HCPs' families, PPE, and mental health resources.

## Supporting information

**S1 Questionnaire. COVID-19 questionnaire English version.**
(DOCX)

**S1 Fig. Predictors of burnout in HICs and LMICs.** Predictors of burnout in HICs (above) and LMICs (below). (PPE) Personal protective equipment.
(DOCX)

**S1 Table. Conversion of ordinal variables into binary.** Conversion of ordinal variables into binary; (QoL) Quality of life; (PA) Physician assistant; (NP) Nurse practitioner; (CRNA) Certified registered nurse anesthetist; (RN) Registered nurse.
(DOCX)

**S2 Table. Country of provenience of study participants and GDP information.** Country of provenience of study participants and classification according to the World Bank.
(DOCX)

**S3 Table. Additional survey responses.** Healthcare professionals responses to perception, exposure, and workload during the COVID-19 pandemic. (HCP) Healthcare professional; (PPE) Personal protective equipment; (QoL) Quality of life; (ICU) Intensive care unit; (ER) Emergency room; (ID) Infectious diseases; (CRNA) Certified registered nurse anesthetist.
(DOCX)

**S1 Data.**
(XLSX)

## Acknowledgments

We acknowledge the support received from UIC Center for Clinical and Translational Sciences' Community Engagement and Collaboration core, Dr. Craig Niederberger and Dr. Ervin Kocjancic. The authors acknowledge the Research Open Access Publishing (ROAAP) Fund of the University of Illinois at Chicago for financial support towards the open access publishing fee for this article.

## Author Contributions

**Conceptualization:** Luca A. Morgantini, Simone Francavilla, Ömer Acar, Daniel Moreira, Michael Abern, Stevan M. Weine.

**Data curation:** Luca A. Morgantini, Heng Wang, Jose M. Flores, Hari T. Vigneswaran, Stevan M. Weine.

**Formal analysis:** Heng Wang, Martin Eklund, Stevan M. Weine.

**Investigation:** Luca A. Morgantini, Ushasi Naha, Simone Francavilla, Ömer Acar, Hari T. Vigneswaran.

**Methodology:** Luca A. Morgantini, Daniel Moreira, Michael Abern, Hari T. Vigneswaran, Stevan M. Weine.

**Project administration:** Luca A. Morgantini.

**Resources:** Luca A. Morgantini.

**Supervision:** Simone Crivellaro, Daniel Moreira, Michael Abern, Martin Eklund, Stevan M. Weine.

**Validation:** Luca A. Morgantini, Heng Wang, Jose M. Flores, Martin Eklund, Hari T. Vigneswaran, Stevan M. Weine.

**Visualization:** Luca A. Morgantini, Hari T. Vigneswaran, Stevan M. Weine.

**Writing – original draft:** Luca A. Morgantini, Ushasi Naha, Hari T. Vigneswaran, Stevan M. Weine.

**Writing – review & editing:** Luca A. Morgantini, Ushasi Naha, Ömer Acar, Simone Crivellaro, Daniel Moreira, Michael Abern, Hari T. Vigneswaran, Stevan M. Weine.

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
