## [Decision Letter · Decision Letter 0]

12 Jun 2020

PONE-D-20-13536

Factors Contributing to Healthcare Professional Burnout During the COVID-19 Pandemic:

A Rapid Turnaround Global Survey

PLOS ONE

Dear Dr. Morgantini,

Thank you for submitting your manuscript to PLOS ONE. After careful consideration, we feel that it has merit but does not fully meet PLOS ONE’s publication criteria as it currently stands. Therefore, we invite you to submit a revised version of the manuscript that addresses the points raised during the review process.

Please see "Additional Editor Comments."

We look forward to receiving your revised manuscript.

Kind regards,

Michio Murakami

Academic Editor

PLOS ONE

Journal Requirements:

The project was supported by the National Center for Advancing Translational Sciences, National

Institutes of Health, through Grant UL1TR002003. We acknowledge the support received from

Sandra Morales-Mirque, Dr. Craig Niederberger and Dr. Ervin Kocjancic.

The authors received no specific funding for this work.

3. Please amend the manuscript submission data (via Edit Submission) to include author Heng Wang.

4. Please amend your authorship list in your manuscript file to include author Heng Want

Additional Editor Comments (if provided):

1. As reviewers pointed out, the authors need to expain participants and recruitment method in more details.

2. The authors need to describe the validity and reliablility of the outcome.

3. It is questionable to compare the level of prevalene in this study (51.4%) with the value (40%) in the other study. Were the same measurement outcome used in this study and the cited study?

4. The authors need to describe the limitations in more details. The limitations may include sample selection biases and validity and reliability of outcomes. If possible, the authors are expected to add how the biases potentlay affected the results and how the authors did the efforts to reduce the biases. Plus, the authors need to add that the causalty is not clarified due to a nature of observational study.

5. Although I recommended the authors to incorporate "Reseach in context" into Introduction and Discussion, although PLOS ONE guidelines are quite flexible and it is not an absolute requirement for the authors to remove this section.

Reviewers' comments:

Reviewer's Responses to Questions

**Comments to the Author**

1. Is the manuscript technically sound, and do the data support the conclusions?

Reviewer #1: Partly

Reviewer #2: No

2. Has the statistical analysis been performed appropriately and rigorously? 

Reviewer #1: No

Reviewer #2: Yes

3. Have the authors made all data underlying the findings in their manuscript fully available?

Reviewer #1: Yes

Reviewer #2: Yes

4. Is the manuscript presented in an intelligible fashion and written in standard English?

Reviewer #1: Yes

Reviewer #2: Yes

5. Review Comments to the Author

Reviewer #1: General

• Limited questionnaire, but not much that can be done about this

• Needs multivariate regression, including demographic variables

Abstract

• Geographical spread is impressive

• Method needs to state definition of burnout

Research in Context

• The ‘Evidence before this study’ merely states a search strategy without stating what the established knowledge was before the study was conducted

Introduction

• This is a minor point. The authors state in the first sentence ‘novel coronavirus (COVID-19)’. The name of the novel coronavirus is SARS-CoV-2, whereas the name of the disease is COVID-19. This should be corrected.

Methods

• The scope of the study in terms of geography and translation into 18 languages is impressive

• It is a shame that validated self-report rating scales were not used, rather than 40 questions based on expert opinion

• Were demographic data collected from respondents? This is not mentioned in the Methods.

• I am not familiar with the burnout literature, but having the main outcome as a single question on self-report burnout seems like an unstable measure that would show high intra-individual variability. It would have been more helpful to use a validated tool, e.g. as listed here: https://nam.edu/valid-reliable-survey-instruments-measure-burnout-well-work-related-dimensions/

• Only participants who responded completely were included in the regression were included. This is reasonable, but it is important to assess whether completers differed from non-completers in any important ways.

• Before participants were asked whether they were ‘burned out’ by their work, were they given any definition of being ‘burned out’?

Results

• 2707 valid responses were received. The manuscript should state how many invalid responses were received.

• Bivariate associations were reported between burnout and various other factors. It would be important to see a multivariate analysis to see which factors were independently associated with burnout.

• In particular, I cannot see any attempt made to examine demographic associations with burnout. These need to be included in a regression model, as they might be significant confounders.

Discussion

• The 3rd sentence mentions factors that typically increase the likelihood of HCP burnout. This needs a citation.

• The 2nd paragraph rather overreaches itself by stating that burnout could be ‘prevented or minimised’. Given that burnout is always prevalent at a certain rate, it is highly improbable that it could be prevented. Minimisation of burnout may be possible, but this is not demonstrated by this paper, which does not examine interventions. This must be stated more cautiously.

• Paragraph 3 states that ‘HCPs who worked extensively during the SARS pandemic in Beijing later demonstrated posttraumatic stress symptoms’. Surely this is only some HCPs.

Reviewer #2: This study reports the prevalence and the risk factors of burnout among healthcare professionals from several countries during the COVID-19 pandemic. The topic is extremely important and the study provide several valuable suggestions to prevent healthcare professionals’ burnout. However, the study design has several serious weaknesses, which limit the validity of their results.

1. As for burnout or mental health status of healthcare professionals during the COVID-19 pandemic, several studies have already been published (e.g. Wu et al., 2020 (J Pain Symptom Manage. doi: 10.1016/j.jpainsymman.2020.04.008); Barello et al., 2020 (Psychiatry Res 290:113129. doi: 10.1016/j.psychres.2020.113129)). Furthermore, there are a lot of studies reporting healthcare professionals’ burnout or mental health issues during the other pandemic, disaster, or in non-disaster settings. Authors should explain the necessity of their study, unexplored research gaps, in the introduction.

2. Please give a more detailed description of the methods used to identify potential participants. How to identify social media groups restricted to HCPs, how many groups were identified and recruited? Were there any inclusion criteria for the groups? Is it possible to estimate the number of potential participants to this survey? Response rate? The authors should report how to distribute their questionnaire.

3. According to the questionnaire and Table 4, the respondents include students, administrative staffs, not a healthcare professional. Were these respondents included in the analyses?

4. The representativeness of the respondents to their survey was not maintained. Therefore, discussion on the prevalence of burnout is impossible based on their data.

5. Burnout was assessed using a single item scale translated into several languages. The reliability and validity of the translated scales, as well as those of cutoff point of 5, were not reported. It is impossible to compare the prevalence of burnout measured by this scale between countries using different languages. Furthermore, the authors assessed only emotional exhaustion and not assessed the other two dimensions of burnout (depersonalization and personal accomplishment). To compare the prevalence of emotional exhaustion and that of burnout is meaningless.

6. Please report the number of respondents who were categorized into high vs low-middle income countries. Is it appropriate to divide the 60 countries into these two groups? Did the authors try to analyze their data with the other categorization of countries? There are many differences among countries, such as the phase of the pandemic, its fatality rate, strategies for COVID-19 adopted, medical system, resources, etc. If authors choose the categorization based on the country’s income level, they should explain its importance on their study. Figure 2 shows the great differences between US and Italy or Sweden among high-income countries. Although the data of the other countries were not reported, the high prevalence of burnout in high-income countries might almost stem from the data in US.

Minor comments

7. Page 4, line 7

Burnout is usually assessed in three dimensions: emotional exhaustion, depersonalization, and personal accomplishment. The authors’ definition of burnout seemed uncommon. Please explain the definition of “personal achievement” or provide the references.

8. Page 6, line 16

The results of the comparison between nurses and physicians reported in the text (OR=1.47) were different from those in Figure 2 (OR=1.12, ns).

9. The authors declared they received no specific funding for this work in the Financial Disclosure section (through online system?) but they acknowledged that their project was supported by the National Center for Advancing Translational Sciences, National Institutes of Health. They should also report the role of the funder.

6. PLOS authors have the option to publish the peer review history of their article (what does this mean?). If published, this will include your full peer review and any attached files.

Reviewer #1: Yes: Dr Jonathan P Rogers

Reviewer #2: No

---

## [Author Response · Author response to Decision Letter 0]

28 Jul 2020

We are truly grateful and humbled for the constructive criticism and suggestions we received from the Editor and the Reviewers.

We believe that your contribution has truly improved the quality of our work and we have learnt important lessons from your feedback.

I extend to you the sincere gratitude of all authors.

Best regards,

Luca Morgantini

---

## [Editor Report · Decision Letter 1]

13 Aug 2020

Factors Contributing to Healthcare Professional Burnout During the COVID-19 Pandemic:

A Rapid Turnaround Global Survey

PONE-D-20-13536R1

Dear Dr. Morgantini,

We’re pleased to inform you that your manuscript has been judged scientifically suitable for publication and will be formally accepted for publication once it meets all outstanding technical requirements.

Kind regards,

Michio Murakami

Academic Editor

PLOS ONE
---

## [Editor Report · Acceptance letter]

24 Aug 2020

PONE-D-20-13536R1 

Factors Contributing to Healthcare Professional Burnout During the COVID-19 Pandemic:
A Rapid Turnaround Global Survey 

Dear Dr. Morgantini:

I'm pleased to inform you that your manuscript has been deemed suitable for publication in PLOS ONE. Congratulations! Your manuscript is now with our production department. 

Kind regards, 

on behalf of

Dr. Michio Murakami 

Academic Editor

PLOS ONE